# The Intersection of Trauma and Immunity: Immune Dysfunction Following Hemorrhage

**DOI:** 10.3390/biomedicines12122889

**Published:** 2024-12-19

**Authors:** Nicholas Salvo, Angel M. Charles, Alicia M. Mohr

**Affiliations:** Department of Surgery, Sepsis and Critical Illness Research Center, College of Medicine, University of Florida, 1600 SW Archer Road Box 100108, Gainesville, FL 32610, USA; nicholas.salvo@ufl.edu (N.S.); angel.charles@surgery.ufl.edu (A.M.C.)

**Keywords:** trauma, hemorrhage, immune dysfunction, inflammation, cytokines

## Abstract

Hemorrhagic shock is caused by rapid loss of a significant blood volume, which leads to insufficient blood flow and oxygen delivery to organs and tissues, resulting in severe physiological derangements, organ failure, and death. Physiologic derangements after hemorrhage are due in a large part to the body’s strong inflammatory response, which leads to severe immune dysfunction, and secondary complications such as chronic immunosuppression, increased susceptibility to infection, coagulopathy, multiple organ failure, and unregulated inflammation. Immediate management of hemorrhagic shock includes timely control of the source of bleeding, restoring intravascular volume, preferably with whole blood, and prevention of ischemia and organ failure by optimizing tissue oxygenation. However, currently, there are no clinically effective treatments available that can stabilize the immune response to hemorrhage and reinstate homeostatic conditions. In this review, we will discuss what is known about immunologic dysfunction following hemorrhage and potential therapeutic strategies.

## 1. Introduction

Trauma is the leading cause of death for people aged forty-five and under, resulting in approximately 275,000 deaths in the U.S. and six million deaths globally each year [1,2,3,4]. It is also responsible for a financial burden of more than four hundred billion dollars each year in the U.S. [5]. Despite major advancements in the acute management of hemorrhage including control of bleeding and hemostatic resuscitation, most preventable trauma-related deaths are due to complications of hemorrhage [5]. Circulatory collapse (typically manifesting as shock) is responsible for approximately 40% of acute deaths from trauma, second only to central nervous system dysfunction, which is responsible for 50% of deaths that occur early after trauma [5]. Traumatic hemorrhage can occur internally or externally, and external bleeding is more readily diagnosed and controlled. Internal hemorrhage may require diagnostic imaging such as a Focused assessment with sonography in trauma (FAST) exam or computed tomography (CT) scan to be identified, delaying recognition, and disruption in internal blood supply creates a pathway for bacteria to enter the bloodstream, making sepsis more likely [6]. Following a hemorrhagic shock, a systemic inflammatory response (SIRS) and a compensatory anti-inflammatory response (CARS) occur simultaneously [7,8] (Figure 1). In some cases, SIRS can become overwhelming and fulminant, leading to death. In survivors, the aberrant immunologic response either reverses rapidly to achieve homeostasis or leads to persistent inflammation and immunosuppression [9]. Patients who are not able to restore immune homeostasis after injury become chronically critically ill and are at risk of developing a constellation of secondary complications including multiple organ failure, immunosuppression, and sepsis, also referred to as persistent inflammation, immunosuppression, and catabolism syndrome (PICS), which is responsible for 20% of trauma-related deaths after hospital admission [9]. Extensive effort has gone into studying the systemic immune dysfunction that results from hemorrhage [10,11,12], and great strides have been made in understanding how these mechanisms are responsible for early and late secondary injury and death, yet little progress has been made in the development of effective therapeutic agents that regulate this immune dysfunction, prevent secondary injury, and restore homeostasis. In this review, our aim is to summarize the current literature on the mechanisms of immune dysfunction after hemorrhage and to discuss potential therapeutic strategies for regulating this response and improving patient morbidity and mortality.

## 2. Physiologic Response to Hemorrhage

The body’s immediate response to hemorrhage involves the neural, hormonal, and immune systems, and these responses can lead to extreme physiologic derangements. The body attempts to prioritize the delivery of oxygen to essential organs by activating compensatory mechanisms during hypovolemia. Baroreceptors and chemoreceptors communicate low blood pressure and hypoxic conditions to the peripheral nervous system which signals the cardiac control center located in the brain stem [13]. Norepinephrine triggers vasoconstriction to compensate for reduced volume, blood flow is decreased to distal arterioles to maintain perfusion of vital organs, and respiration increases [14,15]. During acute hemorrhage, the body also increases the release and circulation of inflammatory cytokines and other signaling molecules that are still being discovered and studied [16]. The ensuing inflammation is often severe and has been associated with multiple organ failure [17]. To control cell death from pro-inflammatory pathways, anti-inflammatory pathways are subsequently upregulated [18]. At this stage, balance between inflammation and immunosuppression is difficult to achieve and maintain, and survivors are left with compromised homeostasis.

## 3. Immune Dysfunction Following Hemorrhage

Hemorrhagic shock has a significant impact on the immune system. The response to acute hemorrhage includes a massive deployment of the immune system to clear debris and eliminate foreign pathogens. The early and exaggerated activation of innate immunity, which is widespread in patients with hemorrhagic shock, results in systemic inflammation, cytokine storm, and excessive activation of complement factors and innate immune cells, comprised of type II innate lymphoid cells, CD4+ T cells, natural killer cells, eosinophils, basophils, macrophages, neutrophils, and dendritic cells [19] (Figure 2). Hemorrhage-induced hypoxia and reoxygenation can activate macrophages and other leukocytes [19]. Natively generated distress signals, known as danger-associated molecular patterns (DAMPs), activate the release of tumor necrosis factor (TNF) and interleukin-1beta (IL-1β) [19,20]. DAMPs and pathogen-associated molecular patterns (PAMPs) also activate the hypothalamic–pituitary–adrenal (HPA) axis and the sympathetic nervous system. These cytokines primarily activate local paracrine elements, including macrophages and neutrophils [20]. Cytokine activation of endothelial cells increases vascular permeability and leads to the release of nitric acid, which causes vasodilation in opposition to the vasoconstriction signaled by the nervous system [21]. During the acute phase there can be elevated levels of endothelial dysfunction. Unresolved hypoxia can disrupt ion distribution and intracellular volume by lowering the fluidity of endothelial membranes and changing membrane potential [21]. Despite clinical advances, survivors of hemorrhagic shock have severe complications due to organ reperfusion injury, including 37% with delayed infections and immune dysfunction, and 22% develop multiple organ failure [22].

### 3.1. Innate Immune Dysfunction

Neutrophils often play a larger role in immune dysfunction or disequilibrium. The endothelium releases interleukin-6 (IL-6) and interleukin-8 (IL-8), both of which contribute to neutrophil attraction [19]. Neutrophils cause the release of new cytokines, which recruit more immune cells to sites of trauma, including mass involvement of monocytes. Monocytes, like neutrophils, have phagocytotic properties and help eliminate infectious pathogens in large numbers. Also, this can contribute to a positive feedback loop until the resolution of the stress or infection. In the case of hemorrhage, this may become ineffective and even harmful as neutrophil apoptosis gives the highly concentrated and longer lasting monocytes control over cytokine release [20]. Monocytes release a cytokine called transforming growth factor-beta (TGF-β) that serves as an anti-inflammatory and immunosuppressing mediator in stress resolution [23]. However, the overexpression of immunosuppressant cytokines like TGF-β can lead to a window of vulnerability to infection and sepsis that may be lethal in patients experiencing or recovering from hemorrhagic shock. Moreover, neutrophils have non-specific immune functions, specifically degranulation and activation of catabolic granules that can promote degradation of deleterious cells and dead tissue. In some cases, the extreme concentration of neutrophils and their uncontrolled apoptosis can lead to toxic elements damaging bodily tissues and leading to cell death [22,24]. Following the increased permeability of endothelial membranes, increased neutrophil migration into certain organs has been shown to be a significant factor causing tissue damage in hemorrhagic shock [25].

### 3.2. Adaptive Immune Dysfunction

The adaptive immune system also undergoes changes and deployment in response to hemorrhagic shock, although it is less involved in the acute phase. Leukocytic dendrite cells and macrophages play a significant role in activating T cells [22]. T cells, specifically regulatory T cells, moderate the resolution of inflammation and release interleukin-10 (IL-10) and TGF-β, both of which have been associated with immune suppression [25]. These cytokines are meant to limit the response of neutrophils and other macrophages causing inflammation in the body and transition back to stable immune expression. However, these cytokines can continue to circulate and downregulate innate immune responses after acute hemorrhage, leading to dysfunction of the immune system.

### 3.3. MDSCs

Severe cases of hemorrhagic shock and patients with compounding factors are more likely to experience complication while attempting to transfer back to homeostasis, often including PICS [9]. During PICS, we must understand the role of myeloid-derived suppressor cells (MDSCs) in the resolution of the acute immune response to trauma. These cells are a group of different immune system agents mobilized from immature myeloid cells in the bone marrow [26,27]. MDSCs are incredibly capable of suppressing adaptive immune system functions, especially T cell expression, and are often associated with pathological conditions like cancer [26,27]. While myeloid cells in the bone marrow usually differentiate into beneficial mature forms of granulocytes or monocytes and help support innate immunity, they can also contribute to harmful cycles of immune disorder during conditions of extreme stress of pathology [26,27]. It is theorized that prolonged exposure to MDSCs may be a root cause of PICS [28]. This pathology is part of a larger system of bone marrow dysfunction. Following severe trauma like hemorrhagic shock, bone marrow dysfunction includes the inhibition of blood cell differentiation, decreased growth of progenitor cells, and even atrophy of the stroma [29,30]. Systemic release of IL-6 and other cytokines from the bone marrow during recovery from severe trauma is part of the dysregulation of inflammation and immunity that leads to chronic pathology [31].

### 3.4. Consequences of Immune Dysfunction

Failure of the immune system during and after hemorrhagic shock leads to infection and sepsis which is common and deadly. As discussed above, the acute hemorrhagic shock response often includes increased permeability of endothelial cells which can enable pathogens and bacteria to infect the body more easily [32]. Endothelial barrier breakdown, responsible for capillary leak, tissue hypoperfusion, and vasoplegia, is a mainstay of secondary organ failure. Sympatho-adrenal hyperactivation appears to be a pivotal driver of this condition. The translocation of microbiota from the gut to other tissues in the body through weakened membranes has been known to be a probable cause of sepsis for decades [33]. Translocation of gut bacteria contributes to multiple organ failure [34]. Other contributing factors to multiple organ failure include the impairment of oxygen delivery, tissue perfusion, and hemodynamic stability stemming from extensive blood loss, which can cause general damage and cell death [34]. It is likely that other comorbid factors influence patient outcomes.

## 4. Metabolic Changes After Hemorrhage

Metabolic changes following hemorrhagic shock are common. It has been shown that cellular hypoxia can interfere with efficient expenditure of energy resources while neurohormonal signaling upregulates glycolysis, creating a catabolic and inefficient metabolic cycle [35]. A lack of oxygen for metabolic processes also leads to acidosis. The stress response that characterizes hemorrhagic shock demands high ATP usage to fuel compensatory mechanisms and tissue repair. When readily available glucose stores run out, hypoglycemia is evident, and cells experience impaired metabolism which can be a major cause for concern. Glucose infusion during extended hemorrhagic shock can improve health outcomes through the emergency maintenance of a highly active metabolism [36]. After hemorrhage, the body shifts to fatty acid consumption with triglycerides meeting approximately 50 to 80% of the consumed energy after trauma. [36] Although it is a less utilized source of energy, proteolysis is also common and deleterious [36]. These metabolic alterations have a significant impact immune cell function in this setting; immune cells shift towards a more glycolytic metabolism, characterized by increased glucose uptake and lactate production. This change in metabolism facilitates the cells’ ability to generate energy needed for activation and migration to the injury site. Unfortunately, this metabolic switch can also lead to excessive inflammation and impaired immune function if not properly controlled [37] The severity of trauma, duration of shock, and onset of infection all play a role in the metabolic response. Similar to the immune system, restoring and maintaining a homeostatic balance is crucial during recovery but can be difficult without medical intervention. Malnutrition can occur following severe trauma, and a supply of sufficient nutrient and energy support is important for successful recovery. Advances in the fields of proteomics and metabolomics will provide a better understanding of the complex biochemistry involved in the metabolic response.

## 5. Current Therapeutic Strategies

Complications including inflammation, immune suppression, metabolic dysregulation, and organ dysfunction lead to difficulties with wound healing, increased susceptibility to infection, and an inability to restore hemostasis following hemorrhagic shock. Currently, therapeutic approaches work toward immediately halting further blood loss by stopping active bleeding and initiating volume replacement, stabilizing inflammatory and immunosuppressant pathways, preventing infection, and providing nutrition, electrolytes, and calories as needed. When specifically analyzing immune dysfunction and treatment, several pharmacological interventions and resuscitation methods have been devised to improve health outcomes; however, clinical success has been limited.

Damage control resuscitation (DCR) is an important concept to limit the harm of hypoxia and hypovolemia as quickly as possible following acute hemorrhage. Gradual fluid replacement has been proposed as a superior alternative to rapid replacement [38]. Also called permissive hypotension, the objective of moderating fluid replacement speed is to avoid dislodging blood clots and diluting the blood to the point of coagulopathy [38]. If a patient receives rapid fluid infusions, especially consisting of fluids besides whole blood, the patient runs the risk of having existing blood clots washed away and being incapable of creating new clots if the blood is sufficiently diluted. This can be very harmful to stemming blood loss and preventing further hemorrhage. Whole blood and derived products are recommended in cases where significant volumes have been lost, as the use of alternatives like crystalloids could critically dilute the blood [39]. However, permissive hypotension is not always viable as severe hypovolemia is extremely deadly and requires immediate intervention. Sometimes, massive blood transfusion may be the only chance of survival, in which case transfusion of plasma, platelets, and other blood products should be performed as soon as possible and in higher quantities [38]. Even when the damage of resuscitation is maximally controlled, blood transfusions also include harmful side effects, like heightened risk or infection and decreased immunity.

A recent study out of Japan comprehensively reviewed data on pre-hospital IV crystalloid transfusions performed on shock patients from 2019 to 2021 [40]. The researchers sought to better understand the impact of early crystalloid transfusions in patients of severe trauma. They found that patients who were given pre-hospital crystalloid IV treatments (lactated Ringer’s) did not experience significantly improved cardiac markers like heart rate and respiratory rate, although there was a slight amelioration in systolic blood pressure, which can be attributed to increased intravenous volume [40]. This study corroborates some existing perspectives on crystalloid solutions failing to improve health outcomes, even when administered immediately following the trauma. Whole blood transfusion in pre-hospital settings is likely advantageous; however, whole blood storage presents challenges that make transportation and readiness more difficult. One relevant case study in Norway demonstrates the benefits and limitations of pre-hospital blood transfusion [41]. Many emergency medical service teams switched to whole blood products for pre-hospital transfusions in the last few years and reported positive outcomes [41]. Most of those who have not yet implemented pre-hospital whole blood transfusion expressed plans or desires to but cited challenges such as the availability and storage of whole blood products [41]. Ultimately, addressing the limitations of whole blood products is likely a more promising strategy than crystalloid or colloid fluids going forward.

Stem cells have the potential to modulate both the local and systemic immune responses. Mesenchymal stem cells (MSC) are multipotent and the source of exosomes and extracellular vesicles (EVs). The ability of MSCs to modulate immune responses is mainly attributed to the secretion of EVs, which mediate intracellular communication and have been proven to modulate local and systemic immune responses in animal models [42]. Cell-based therapies using MSCs or MSC-EVs have been shown to be beneficial for improving neurologic outcomes and lung injury in animal models of hemorrhagic shock [42].

An early single dose of exosomes derived from MSCs has been shown to attenuate neurological injury by decreasing IL-1, IL-6, and IL-18 and increasing granulocyte–macrophage colony-stimulating factor (GM-CSF) levels in a swine model of hemorrhagic shock [39]. Following injury, the concentrations of EVs increase in the blood, supporting the concept that they participate in physiological responses to hemorrhagic shock, including endothelial damage, heightened inflammation, and tissue damage; interestingly, these EVs were primarily derived from endothelial cells, platelets, and leukocytes [43]. However, survival of hemorrhagic shock is correlated with higher EV levels, whereas decreased levels often led to higher transfusion requirements and worse outcomes [43]. It is also theorized that EVs play a key role in contributing to blood clot formation. Studies have found that the removal of EVs from the blood leads to decreased thrombin generation and that the restoration of EV levels contributes to thrombin regeneration [42]. Critically, blood products, specifically plasma, which have been frozen or stored for multiple days have lowered EV counts and could hinder blood clot formation in patients who must halt continued bleeding and need massive blood transfusion [41]. EVs engage with complex molecular pathways and all their interactions and contributions are not yet understood. New research has experimentally shown that platelet-derived EVs (PDEVs) moderate ischemia and help against metabolic acidosis [43]. As further study elevates our appreciation for the benefits of EVs and platelet-derived EVs in the blood, and in blood transfusion, targeted deployment of prepared EVs may emerge as an effective treatment to help promote coagulation and balanced recovery. For now, these studies illustrate the need for fresh blood products to be readily available for DCR.

Efforts to help modulate the immune system and prevent overreactive inflammation include many options for pharmaceutical intervention with varying rates of success. Corticosteroids are often used in the treatment of septic shock to limit inflammation causing organ dysfunction [44]. Certain corticosteroids, such as hydrocortisone and fludrocortisone, have been shown to significantly reduce mortality and shorten recovery time at low doses [45]. However, larger, more recent studies performed on the same corticosteroids did not find a significant reduction in mortality, although the shortened recovery time was supported [45]. It is important to note that corticosteroids at low doses are still generally thought of as safe and may help stabilize patients more quickly even if they do not directly prevent multiple organ dysfunction syndrome (MODS) or multiple organ failure (MOF). Another avenue of potentially beneficial immunomodulation is through the inhibition of inflammation mediating interleukins, specifically IL-1 and IL-18. Studies on mice have found that using IL-18 antibodies and IL-1 receptor antagonists in combination have prevented mortality from TNF-induced septic shock [46]. An experimental study on human patients failed to significantly reduce mortality through inhibition of IL-1 [46]. A possible explanation is that in uncontrolled environments patients may be at distinct stages in the inflammatory pathway and inhibition of IL-1 is more effective immediately following trauma as opposed to even just hours later, as IL-1 acts mostly as an early cytokine which activates downstream agents [47]. This is supported by the fact that IL-1 concentrations are only elevated in the immediate response to trauma and quickly decrease to base levels [48]. Inhibition of inflammatory cytokines is likely to be a fruitful field of study and treatment even if mechanisms of intervention are extremely complicated [49,50].

While some pharmaceutical interventions seek to reduce inflammation considering the damage cycles that extreme inflammation can have on tissues and organ systems, other interventions seek to bolster the immune system in recognition of the role that infection can play in sepsis following hemorrhagic shock. These goals may be difficult to simultaneously achieve as countering inflammation often inadvertently suppresses the immune system, further enabling infection that may also cause or help contribute to MOF [51]. Thymosin alpha-1 (TA1) is a peptide hormone which bolsters immune activity through the activation of T cells and dendrites and regulates cytokine production [52]. While it has long been used in the treatment of hepatitis, there is high potential for the use of TA1 in immunocompromised patients [53,54]. TA1 has also been proven to be effective in cases of septic shock in combination with other medical treatments [55]. The same study also noted that some patients experience an overactive immune system and hyperinflammation while others with similar trauma experienced immunosuppression [56,57]. This makes the re-establishment of homeostasis difficult. There is hope that TA1 or a similar immune regulator may help balance cytokine levels during recovery.

Emerging evidence has also shown that microRNAs (miR), a subset of short single-chain, non-coding RNAs that regulate gene expression, play an essential role in the pathophysiologic response to hemorrhagic shock by regulating inflammation and immunity. For instance, knockdown of miR-18b was shown to reduce the levels of superoxide dismutase, inducible nitric oxide synthase, and IL-6 in macrophages. In addition, a reduction of miR-18b decreased the M1/M2 ratio of macrophages and reduced the Th1/Th2 ratio of CD4+ T cells in splenic tissues after hemorrhagic shock injury [58]. This shift towards a Th2 dominant response is characterized by an increased production of Th2 cytokines like IL-4 and IL-5, which is associated with inflammation and impaired wound healing [59]. Moreover, in a preclinical model of trauma hemorrhagic shock, 44% of miR were significantly upregulated compared with control mice. In particular, miR-638, miR-135a-5p, miR-135b-5p, miR-668-3p, miR-204-5p, miR-146a-5p, miR-200a-3p, miR-17-5p, miR-30a-5p, and miR-214-3p were found positively correlated with lactate and negatively with base excess and bicarbonate, clinical parameters that reflect the severity of shock [60]. Importantly, although our understanding of the immune response to trauma and hemorrhagic shock continues to expand, these exciting discoveries have not been successfully developed into effective treatment strategies, and standard of care remains supportive.

## 6. Conclusions

Despite extensive investigation, there are still major gaps in our current understanding of what drives immune dysfunction after hemorrhagic shock or how to restore homeostasis. The regulation of the immune response to hemorrhagic shock is multifactorial and quite complex, making the development of effective therapeutics difficult. Researchers continue to map pathways and better understand the relationships between different agents. Due to the interconnected and compensatory nature of the hormonal, neural, immune, and metabolic systems, some interventions have not only proven to be ineffective, but have also caused unintended and even opposite consequences. Additionally, attempts to treat specific phases of inflammation or over/under expression of certain molecules following hemorrhagic shock may, in fact, be targeting the wrong step or agent. It is critical that the immune response is reliably monitored temporally before any therapeutic immunomodulation can be performed effectively. Because innate and adaptive immune responses can differ depending on age, comorbidities, and other preexisting conditions, efforts to delineate the underlying posttraumatic mechanisms need to be intensified. In the era of precision medicine, big data-driven discovery in complex trauma situations might be feasible using bioinformatics tools which could improve the phenotyping of injury patterns, precision diagnosing, and treatment.

In future directions, it is crucial that animal studies evaluate the impact of different immunomodulators and measure their molecular consequences very closely. Experimental studies must corroborate any proposed pharmaceutical intervention. We should be able to compare the short-term and long-term effects of every treatment and analyze the side effects, mortality rates, and recovery times, keeping in mind that it is possible for no intervention to have been better. The future might thus yield insights into multiple common and interactive immune responses, including immuno-metabolic and neuro-immunological switches and checkpoints after trauma. Overall, therapeutic immunomodulation may not only aim to stabilize fluid-phase and cellular innate immunity along with immunological barriers but also use ex vivo reprogrammed cells to induce regenerative processes and to promote healing and improve long-term outcomes following trauma.

In conclusion, hemorrhagic shock is an extreme and deadly condition, the survival of which depends upon urgent intervention and cessation of blood loss, which is followed by DCR, and modulating the subsequent immune response. The sequela of hemorrhagic shock is defined by cycles of inflammation and immunosuppression which can harm native tissues and leave the body vulnerable to infection. Widespread alteration to metabolic and hormonal systems leaves homeostasis compromised. MODS leading to MOF is unfortunately common during recovery from hemorrhagic shock. Pharmaceutical interventions seek to limit inflammation, support the immune system, or regulate the cascades of cytokines that dominate these spikes in the hopes of restoring homeostasis.

## Figures and Tables

**Figure 1 biomedicines-12-02889-f001:**
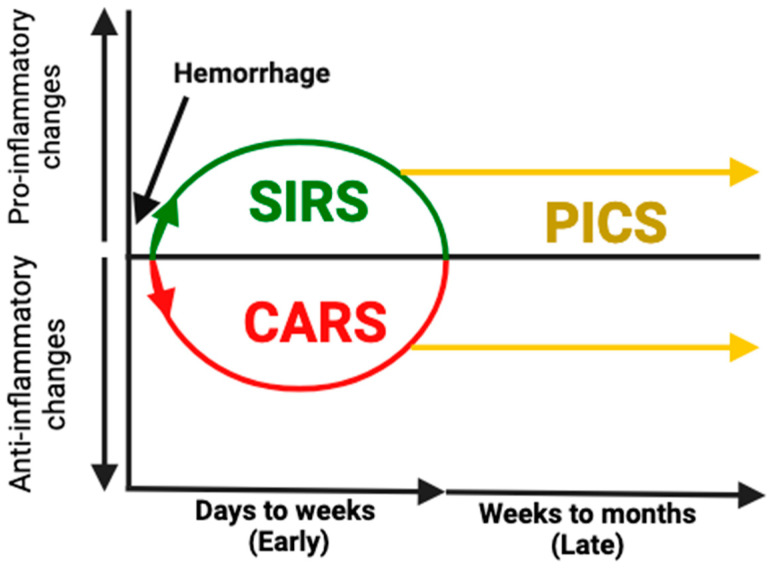
Acute and chronic immune changes after hemorrhage. After hemorrhage, a hyperinflammatory response known as the systemic inflammatory response syndrome (SIRS) occurs. This response triggers an immune suppressing response known as the compensatory anti-inflammatory response syndrome (CARS). These changes occur early after injury and lead to multiorgan dysfunction and early infections. For many patients, these responses resolve, and immune homeostasis is restored; however, in many patients, homeostasis is never restored, and they develop persistent inflammatory–immunosuppressive and catabolic syndrome (PICS). PICS can persist for months, leading to persistent organ dysfunction, opportunistic infections, and death.

**Figure 2 biomedicines-12-02889-f002:**
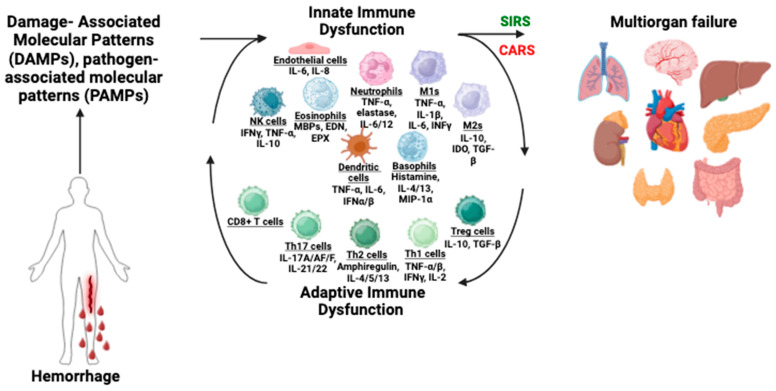
Innate and adaptive immune dysfunction after hemorrhage leads to multiorgan failure. Hemorrhage causes the release of damage-associated molecular patterns (DAMPs) and pathogen-associated molecular patterns (PAMPs), which causes early and excessive activation of the endothelial system, complement factors (not depicted here), and immune cells including macrophages (M1 and M2), dendritic cells (DC), T cells, natural killer (NK) cells, eosinophils, basophils, and neutrophils. These activated cells secrete cytokines and chemokines, which exacerbate inflammation and subsequent immunosuppression. The result of these pathologic immune changes is the development of systemic immune response syndrome (SIRS) and a compensatory anti-inflammatory response syndrome (CARS), which ultimately lead to multiple organ failure. Abbreviations: IL—interleukin, IFN—interferon, TNF—tumor necrosis factor, MBP—major basic protein, EDN—eosinophil derived neurotoxin, EPX—eosinophil peroxidase.

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
