# Peer review of "The Intersection of Trauma and Immunity: Immune Dysfunction Following Hemorrhage"

_biomedicines, 2024, doi:10.3390/biomedicines12122889_

Round 1

Reviewer 1 Report

Comments and Suggestions for Authors

The review is of great interest to specialists in the field of molecular biology and medicine. The article is well written and divided into sections. It is easy to read. The article «The Intersection of Trauma and Immunity: Immune Dysfunction Following Hemorrhage», by Nicholas Salvo, Angel M. Charles, and Alicia M. Mohr can be published in the journal Biomedicines. However, there are some comments I have to make.

COMMENTS

1.         As far as I can judge, hemorrhage after trauma can be both internal and external. I think this should be emphasized in the Introduction. Most likely, sepsis is mainly caused by internal hemorrhage.

2.         Line 32

Does «early death from trauma» means death of young (under 45 years) patients? Please explain in the text the concept of «early deaths».

3.         Line 34

Please explain in the text the concept of «early trauma»

4.         Line 38

Do the authors include those who develop persistent inflammation and immunosuppression in the "chronic critically ill patients"? Please clarify this in the text.

5.         Lines 38-47

These chronic critically ill patients are at risk of developing a constellation of secondary complications including multiple organ failure, immunosuppression, and sepsis, also referred to as Persistent Inflammation, Immunosuppression, and Catabolism Syndrome (PICS), which is responsible for 20% of trauma-related deaths after hospital admission [8] Extensive effort has gone into studying the systemic immune dysfunction that results from hemorrhage [9-11], and great strides have been made in understanding how these mechanisms are responsible for early and late secondary injury and death, yet little progress has been made in the development of effective therapeutic agents that regulate this immune dysfunction, prevent secondary injury, and restore homeostasis.

These are probably two different sentences. Please, separate them.

6.         Figure 2.

There are many abbreviations in Figure 2 that need to be explained not only in the text, but also in the figure legend. It is necessary to replace “TGF-B” with “TGF-β”, “IL-1b” with “IL-1β”, etc. in the figure.

7.         Line 125-127

Following the increased permeability of endothelial membranes, increased neutrophil migration into certain organs has been shown to be a significant factor causing tissue damage in hemorrhagic shock.

Some references to the original works are required.

8.         Lines 176-178

«The body shifts to fatty acid consumption, if possible, although proteolysis is also common and deleterious [36].»

Does the phenomenon described occur as a result of glucose infusion? Please explain in the text.

9.         Line 214

«Cell-based therapies using MSCs or MSC-EVs»

Please explain in the text the concept of «MSC-EVs».

10.       Line 220-221

«Following injury, the concentrations of distinct types of EVs increase in the blood»

Please explain in the text what types of EVs can present in the blood.

11.       Line 247

You need to introduce the terms MODS and MOF in the text.

12.       Line 280

Are other microRNAs besides miR-18 known to influence hemorrhagic shock? If there are data on other small RNAs, this should be indicated in the text.

13.       Line 281

Please explain in the text the concept of «Th1/Th2 ratio».

14.       Lines 293-294

You need to remove the gap between the lines.

15. Ref 25

Please, in ref. 25 add page numbers

16. The Review should include an analysis of articles published in the last years (2023 and 2024).

Author Response

1. Summary

Comment 1: As far as I can judge, hemorrhage after trauma can be both internal and external. I think this should be emphasized in the Introduction. Most likely, sepsis is mainly caused by internal hemorrhage.

Response 1: Thank you for pointing this out. We agree with this comment. Therefore, we have discussed this point in the introduction. Please see lines 34-39.

Comment 2: Line 32. Does «early death from trauma» means death of young (under 45 years) patients? Please explain in the text the concept of «early deaths».

Response 2: Thank you for pointing this out. We agree with this comment. Therefore, we have clarified. Please see line 32, 34.

Comment 3: Line 34. Please explain in the text the concept of «early trauma»

Response 3: Thank you for pointing this out. We meant early as in acute or occurring shortly after injury.

Comment 4: Line 38. Do the authors include those who develop persistent inflammation and immunosuppression in the "chronic critically ill patients"? Please clarify this in the text.

Response 4: Thank you for pointing this out. Please see line 44-45.

Comment 5: Lines 38-47. These chronic critically ill patients are at risk of developing a constellation of secondary complications including multiple organ failure, immunosuppression, and sepsis, also referred to as Persistent Inflammation, Immunosuppression, and Catabolism Syndrome (PICS), which is responsible for 20% of trauma-related deaths after hospital admission [8] Extensive effort has gone into studying the systemic immune dysfunction that results from hemorrhage [9-11], and great strides have been made in understanding how these mechanisms are responsible for early and late secondary injury and death, yet little progress has been made in the development of effective therapeutic agents that regulate this immune dysfunction, prevent secondary injury, and restore homeostasis.

These are probably two different sentences. Please, separate them.

Response 5: Thank you for pointing this out. We separated the sentences.

Comment 6: Figure 2. There are many abbreviations in Figure 2 that need to be explained not only in the text, but also in the figure legend. It is necessary to replace “TGF-B” with “TGF-β”, “IL-1b” with “IL-1β”, etc. in the figure.

Response 6: The recommended changes were made to the figure.

Comment 7: Line 125-127. Following the increased permeability of endothelial membranes, increased neutrophil migration into certain organs has been shown to be a significant factor causing tissue damage in hemorrhagic shock. Some references to the original works are required.

Response 7: References were added.

Comment 8: Lines 176-178. «The body shifts to fatty acid consumption, if possible, although proteolysis is also common and deleterious [36].»

Does the phenomenon described occur as a result of glucose infusion? Please explain in the text.

Response 8: Please see lines 191-200.

Comment 9: Line 214. «Cell-based therapies using MSCs or MSC-EVs». Please explain in the text the concept of «MSC-EVs».

Response 9: Please see lines 236-238.

Comment 10: Line 220-221. «Following injury, the concentrations of distinct types of EVs increase in the blood». Please explain in the text what types of EVs can present in the blood.

Response 10: Please see lines 246-247.

Comment 11: Line 247. You need to introduce the terms MODS and MOF in the text.

Response 11: Please see line 272.

Comment 12: Line 280. Are other microRNAs besides miR-18 known to influence hemorrhagic shock? If there are data on other small RNAs, this should be indicated in the text.

Response 12: Please see lines 309-316.

Comment 13: Line 281. Please explain in the text the concept of «Th1/Th2 ratio».

Response 13: Please see lines 306-308.

Comment 14: Lines 293-294.You need to remove the gap between the lines.

Response 14: The gap was removed.

Comment 15: Ref 25. Please, in ref. 25 add page numbers

Response 15: Page numbers were added.

Comment 16: The Review should include an analysis of articles published in the last years (2023 and 2024).

Response 16: More recent articles were added. 

Reviewer 2 Report

Comments and Suggestions for Authors

The authors present a review of immune dysfunction following hemorrhage. 

What is provided is a very general overview of the topic, which would benefit greatly by being more concise in its language for the current content, and supplemented by further detail and logical additional sections.

Examples of this would include division of section 3 into affected subsets, and the delineation between subsets affecting and being affected. 

Section 4 doesn't delve into the effects of the stated metabolic changes to immune cells.

Section 5 tentatively highlights therapeutic strategies, most of which do not actually work. The authors' opinions for the potential of these strategies in this section are too much of a stretch and not really appropriate.

The conclusions would benefit from being more clearly written. Also, the sentence beginning on line 298 needs to be removed. Promoting animal studies in this way is very contentious today.

Comments on the Quality of English Language

Please make the writing more formal. The language used throughout is too simplistic.

Author Response

1. Summary

Comment 1: What is provided is a very general overview of the topic, which would benefit greatly by being more concise in its language for the current content, and supplemented by further detail and logical additional sections.

Response 1: Thank you for your feedback. We have made changes to the language throughout. They are highlighted in yellow.

Comment 2: Examples of this would include division of section 3 into affected subsets, and the delineation between subsets affecting and being affected. 

Response 2: Thank you for your feedback. We have divided section 3 into subsets and clarified whether the immune sections described are consequences or causes.

Comment 3: Section 4 doesn't delve into the effects of the stated metabolic changes to immune cells.

Response 3: Thank you. Please see lines 194-200.

Comment 4: Section 5 tentatively highlights therapeutic strategies, most of which do not actually work. The authors' opinions for the potential of these strategies in this section are too much of a stretch and not really appropriate.

Response 4: Thank you. We agree and we altered this section to make it more objective and highlighted the fact that many of these therapies are ineffective.

Comment 5: The conclusions would benefit from being more clearly written. Also, the sentence beginning on line 298 needs to be removed. Promoting animal studies in this way is very contentious today.

Response 5: Please see changes to the conclusion in lines 318- 325.

Comment 6: Please make the writing more formal. The language used throughout is too simplistic.

Response 6: Thank you for the feedback. We have made changes to the language.

Round 2

Reviewer 1 Report

Comments and Suggestions for Authors

The authors of the article took most of the comments into account. The article «The Intersection of Trauma and Immunity: Immune Dysfunction Following Hemorrhage», by Nicholas Salvo, Angel M. Charles, and Alicia M. Mohr can be published in the journal Biomedicines with corrections.

6. Not all terms in the Figure 2 are corrected: TNF-a/b (Th1 cells), IFNa/b (Dendritic cells), IFNy (IFNγ, NK cells and Th1 cells), etc.

Legend to Fig.2. «IL=interleukin» must be replaced by «IL-interleukin», etc.

12. Line 309. «MiR-638» must be replaced by «miR-638»

16. Although the authors have added an analysis of the 2023 article, however, from my point of view, the authors do not fully present the latest data on the stated topic of the review. I strongly recommend to add other articles from recent years.

Author Response

Comment 1: Not all terms in the Figure 2 are corrected: TNF-a/b (Th1 cells), IFNa/b (Dendritic cells), IFNy (IFNγ, NK cells and Th1 cells), etc.Legend to Fig.2. «IL=interleukin» must be replaced by «IL-interleukin», etc.

Response 1: The terms in the figure and it's legend were updated.

Comment 2: Line 309. «MiR-638» must be replaced by «miR-638»

Response 2: MiR was changed to miR.

Comment 3: Although the authors have added an analysis of the 2023 article, however, from my point of view, the authors do not fully present the latest data on the stated topic of the review. I strongly recommend to add other articles from recent years.

Response 3: Thank you for your insight. We did another literature review and found a few more recent resources that we think capture what is currently understood about the topic more thoroughly.

Reviewer 2 Report

Comments and Suggestions for Authors

The authors have modified the text to a satisfactory degree following the suggestions from peer review. 

Author Response

Comment 1: The authors have modified the text to a satisfactory degree following the suggestions from peer review. 

Response 1: Thank you very much for your feedback.
